# Active Surveillance of Root Caries in Vivo with CP-OCT

**DOI:** 10.3390/diagnostics13030465

**Published:** 2023-01-27

**Authors:** Yihua Zhu, Minyoung Kim, Donald Curtis, Jing Wang, Oanh Le, Daniel Fried

**Affiliations:** Department of Preventive and Restorative Dental Sciences, University of California, San Francisco, 707 Parnassus Ave, San Francisco, CA 9414, USA

**Keywords:** root caries, optical coherence tomography, lesion activity, non-surgical intervention

## Abstract

The active surveillance of root caries lesions to monitor potential remineralization or decay progression is challenging for the clinician, due to unreliable diagnostic information. The conventional visual and tactile methods for assessing the lesion activity are not reliable, and the clinician is often unable to determine if the lesion is progressing or has been arrested. An important marker of an arrested lesion is a highly mineralized transparent surface zone (TSL) that forms when the mineral is deposited in the outer layer of the lesion. The purpose of this study was to determine if cross-polarization optical coherence tomography (CP-OCT) could be used to detect changes in the lesion severity and activity during active monitoring. In total, 18 subjects with 22 suspected active root caries lesions were evaluated using CP-OCT at the baseline, 3 months, and 6 months. All subjects were instructed to use a high fluoride dentifrice at the baseline. The results showed that CP-OCT was able to discriminate the active from the arrested lesions by identifying the presence of a TSL on arrested lesions. The results also indicated that the mean TSL thickness increased significantly (*p* < 0.05) for the nine lesion areas. In addition, CP-OCT was able to show the progression of demineralization, erosion, and changes in gingival contours in scanned areas. CP-OCT was valuable for monitoring the activity and severity of root caries lesions in vivo. CP-OCT can be used to assess the activity of root caries lesions at a single time point by detecting the presence of a TSL at the lesion surface indicative of the lesion arrest.

## 1. Introduction

Root caries remains a clinical concern, especially in the elderly [1]. Their management remains challenging because the clinical diagnosis of root caries is subjective and is based on visual and tactile parameters. In contrast to coronal caries, root caries lack a valid diagnostic standard, such as radiography [2]. Investigators have not developed a reliable relationship between root caries’ appearance and activity [3,4]. Most experts agree that active root lesions are soft, yet tactile hardness assessments remain subjective and lack reliability [3]. Multifactorial root caries scoring systems have been developed with mixed success [5,6]. The International Caries Detection and Assessment System (ICDAS) coordinating committee proposed a system for assessing lesion activity [5,7]. Criteria include color; texture (smooth, rough); appearance (shiny, dull); tactile (soft, leathery, hard); loss of anatomical contour; and proximity to the gingival margin [8,9]. Histological analyses for lesion assessment, such as microradiography, requires the destruction of the tooth and are not suitable for use in vivo. Optical coherence tomography (OCT) is a noninvasive technique for creating cross-sectional images of the internal biological structure and is well-suited for imaging coronal and root surfaces in vivo [10,11,12]. Polarization sensitivity has proven to be valuable for enhancing the contrast of demineralization, reducing the strong specular reflection from tooth surfaces, and measuring birefringence [13,14,15]. Studies have demonstrated that CP-OCT can be used to quantify the loss of cementum and the severity of the demineralization on root surfaces [16,17]. OCT can be used to discriminate between non-carious cervical lesions and root caries in vivo [18]. Importantly, CP-OCT can be used to monitor the formation of a highly mineralized transparent surface layer (TSL) on demineralized dentin after exposure to a remineralization solution [17,19]. The optical changes associated with the lesion dehydration have been investigated via thermal, fluorescence, and SWIR imaging for lesions on enamel surfaces [20,21,22,23,24]. Thermal imaging during the lesion dehydration has been used to measure the lesion permeability [19,25,26,27]. Decreased thermal emission from the lesion surface is indicative of reduced permeability and a drying rate that is consistent with an arrested lesion. During drying, lesion areas will cool due to the loss of water from porous areas and from the collapsed and unsupported collagen matrix of the demineralized cementum and dentin. Arrested lesions have mineral filling the pores at the lesion surface due to the remineralization and the formation of a TSL, which inhibits water diffusion and permeability, resulting in less temperature change and shrinkage during drying. Lesions with a TSL have significantly lower thermal emission [19,25,26,27,28].

In a recent study, CP-OCT and thermal imaging were used to assess the activity of root caries lesions on 30 test subjects [27]. In that study, CP-OCT was used to measure the lesion structure and the presence of TSL formation indicative of arrested lesions at a single time point. The activity of those lesions was confirmed with thermal imaging. The next step is to consider if CP-OCT can detect changes in the lesion activity over time during active monitoring. The purpose of this study was to monitor root caries lesions over a period of 6 months to determine if CP-OCT could be used to detect changes in lesion severity and activity. Subjects with root caries lesions that were clinically diagnosed as active were prescribed a high fluoride dentifrice and monitored from baseline to 6 months.

## 2. Materials and Methods

### 2.1. Participant Recruitment and Procedures

Thirty test subjects at the UCSF comprehensive clinics were recruited for this study with approval from the Human Protection Program Institutional Review Board of the University of California, San Francisco (IRB#18-24558, Approval Date: 27 March 2018). This study was a single-site longitudinal observational clinical study. Test subjects were aged 18–90 and had one root caries lesion that was diagnosed as active based on a visual and tactile assessment by an experienced clinician using the ICDAS II root caries lesion activity criteria [7]. Test subjects were prescribed and instructed to use a high fluoride (5000 ppm) dentifrice, Prevident 5000 Plus (Colgate, NY, USA). Test subjects were imaged at baseline, 3 months, and 6 months with CP-OCT after beginning brushing with a high fluoride toothpaste. Color images were acquired at the baseline and 6 months using a FocusDent MD740 (Vilnius, Lithuania) 1280 × 960-pixel intraoral camera. The sample size of 30 was based on an in vitro study of the remineralization of simulated dentin lesions, with the mean transparent surface layer (TSL) thickness increased from 0 to 53.8 ± 13.3 µm after 12 days of remineralization [19]. Therefore, based on an average SD of 6.65 µm for the TSL thickness with a two-sided significance level of 0.05, a study with 30 subjects would have a 90% power to detect an increase in the TSL thickness of 5.7 µm.

### 2.2. Cross-Polarization Optical Coherence Tomography (CP-OCT)

The cross-polarization OCT system used for this study was the Model IVS-3000-CP, purchased from Santec (Komaki, Aichi, Japan). This system acquires only the cross-polarization image (CP-OCT), not the simultaneous cross and co-polarization images (PS-OCT). This is a 3D system with a right-angle dental handpiece capable of acquiring complete tomographic images of a volume of 6 × 6 × 7 mm in approximately 3 s. This system has been used for multiple in vivo caries imaging studies [29,30,31]. It operates at a wavelength of 1321 nm and a bandwidth of 111 nm, with a measured resolution in the air of 11.4 µm.

An attachment with an internal air nozzle made of autoclavable Dental-SG resin was printed using a Formlabs 3D printer (Somerville, MA, USA) and attached to the OCT scanning handpiece. The air at 10 psi was connected to the probe to prevent the fogging of the imaging window and to dry the tooth surface. Images were acquired after drying for 30 s. Images of the CP-OCT handpiece are shown with and without the 3D printed attachment in Figure 1. The entire handpiece assembly was covered with polypropylene film for infection control prior to the imaging, as shown in Figure 1. Entire (3D) CP-OCT cross-polarization images of each lesion were acquired at each of the three timepoints. Image preprocessing was applied using ORS Dragonfly (Montreal, QC, Canada) to despeckle and smooth the images before image registration [32,33]. Image segmentation was performed with 90% intensity thresholding, followed by the application of a 3D median filter with a kernel size 3 for noise reduction and smoothing, while preserving the potential transparent surface layer (TSL). Image registration was first performed using manual rotation and translation to achieve an initial rough alignment. It was followed by the application of ORS Dragonfly’s automatic image registration function. It automatically tests the registration by introducing miniature steps of rotation as small as 0.5 degrees and the translation of 1 voxel with a linear type of interpolation until the mobile image is superimposed to the fixed image. The alignment is assessed by the sum of squared differences (SSD) method. It is important to note that OCT measures the optical path length and not the physical path length, so some distortion is expected for images acquired at different angles.

For quantitative measurements, rulers with a diameter of 100 μm were created in ORS Dragonfly representing the respective intensity profiles orthogonal to the tooth surface at positions in each lesion area. Positive TSL detection was confirmed with two significant consecutive intensity peaks detected at the beginning of the ruler, and the TSL thickness was measured as the distance between the two peaks. The lesion depth (Ld) was measured as the overall length of the ruler’s non-zero intensity profile. To prepare for the integrated reflectivity (ΔR) measurement, an intensity histogram of the cylindrical ruler was exported from Dragonfly, and analyzed with a custom script written with MATLAB from Mathworks (Natick, MA, USA) to identify the median intensity over the lesion depth that was later used as an intensity normalization correction factor. Each OCT scan has a different intensity profile due to the variation in the angle and position of the imaging and other environmental factors, so normalization was necessary for accurate ΔR comparison across all three co-registered images. Furthermore, ΔR was then calculated as the sum of the intensity over the lesion depth divided by the normalization correction factor.

## 3. Results

In this study, only 22 of the 30 test subjects initially recruited completed all three visits. The COVID-19 pandemic greatly interfered with the study. Test subject visits at UCSF for clinical studies considered nonessential for patient health were banned during part of the pandemic. Images from 18 of the test subjects were successfully co-registered, while data from 4 of the test subjects were discarded due to the inability to co-register the images at the baseline, 3 months, and 6 months. Four subjects presented with both clinically appearing active and arrested caries on the same tooth, based on the presence of a transparent surface zone (TSL), so that twenty-two lesion areas were included for analysis from the 18 test subjects. Eight of the lesion sites were arrested at the baseline, as shown by a visible TSL, and only one active lesion became arrested during the 6 months of the study, according to CP-OCT. Therefore, 9 of the 22 lesion sites had a measurable TSL at month 6, indicating those lesion areas were arrested. Interestingly, many of the arrested lesions showed an increase in the TSL thickness from baseline to 6 months. All the lesions were initially classified as active lesions based on the clinical assessment and the multiple ICDAS II lesion activity criteria. Of the 22 lesion sites analyzed, 8 were dark and 14 were light-colored at the baseline, and 10 appeared rough and 12 appeared smooth. Digital images of the lesions at the baseline and 6 months were shown to two clinicians after the completion of the study, and they were asked to score the caries status and the activity of the lesions at the two time points based on digital photographs. Table 1 is a contingency table comparing the caries status (caries present or not present) and caries activity (active and arrested) assessed by the visual examination of the images with CP-OCT assessment, in which the presence of a TSL indicated the lesion was arrested, while the presence of demineralization without a TSL indicated an active lesion. CP-OCT indicated that demineralization was present for all lesion sites confirming root caries presence.

At the baseline, 9 of the 14 active lesions were visually classified correctly, while none of the 8 arrested lesions were correctly visually classified as arrested. At month 6, only 5 of the 13 active lesions were visually classified correctly, while none of the 9 arrested lesions were correctly visually classified as arrested. The two lesions that were classified as arrested by the clinicians at month 6 were shown to be active in CP-OCT images at month 6.

An example of an arrested lesion with minimal change after 6 months is shown in the color and CP-OCT images of Figure 2. In the color images, the lesion area looks similar, however, the exposed root surface appears to have decreased due to growth of the gingival margin. A surface rendering (3D) of the entire CP-OCT scan taken at the baseline is also shown, with the yellow line showing the position of the extracted CP-OCT b-scans. The CP-OCT scan at the baseline shows the gingiva on the left and the exposed root surface on the right. The lesion appears with a lesion body of higher intensity covered by a thin transparent layer (dark zone) that indicates the TSL. The TSL has a very thin layer of higher intensity that represents reflectance from the tooth surface. The lesion does not appear to change in depth or severity after 6 months.

A second lesion that is active with advancing cavitation is shown in Figure 3. In the color image, a shadow is visible just above the exposed lesion area at the baseline and that area is no longer visible at month 6. It is hard to determine from the color images whether or not the lesion area is larger. The surface renderings of the CP-OCT scans that are shown at the baseline and 6 months show that the area of cavitation is larger, and that the area near the top of the lesion has collapsed. The CP-OCT scan at the baseline extracted at the position of the yellow line shows the gingiva on the left and a cavitated lesion area on the right, along with the mostly transparent sound enamel. A small lesion area (LB) is visible just below the enamel, demarcated by the orange arrows. That lesion area between the orange arrows becomes cavitated after 6 months in the lesion area. The depth lost was measured to be 151 µm from 0–3 months and 296 µm from 3–6 months.

A third representative lesion that appears initially active at the baseline and appears to become arrested at month 6 is shown in Figure 4. The color images show little change after 6 months. The CP-OCT b-scan at the baseline extracted at the position of the yellow line in the CP-OCT surface rendering shows a highly reflective, slightly cavitated, active lesion located between the enamel and the gingiva. In this scan, the gingiva is on the right and the enamel is on the left. The lesion also extends into the enamel above the exposed root surface. Between the baseline and month 6, the reflectivity of the lesion body decreases from both the enamel and dentin lesion (exposed root) areas, and a distinct TSL forms over the lesion.

A fourth lesion is shown in Figure 5. Color images show a large area of the exposed root surface at baseline that becomes almost completely covered by inflamed gingiva after 6 months. CP-OCT surface renderings also shown at the baseline and 6 months show dramatic hypertrophy of the gingival tissues. CP-OCT b-scans extracted at the position of the yellow line show that the tooth is covered by a crown, and that there is an exposed cavitated area of the root surface located between the gingival (left) and the crown (CR) on the right, located between the two orange areas. The CP-OCT scan also shows that there is some shallow demineralization on the exposed root surface, but it is localized and very shallow. The month 3 and 6 CP-OCT scans show that the gingiva expands to cover much of the exposed root surface, and that the lesion is much smaller at month 6. The lesion depth, integrated reflectivity over the lesion depth (ΔR), and the thickness of the transparent surface zone if present (TSL) were measured from the co-registered CP-OCT scans. Values were calculated for 22 lesion areas from 18 teeth, and the mean ± S.D. are plotted for all three values in Figure 6. Both the mean lesion depth and the mean ΔR increased slightly from the baseline to month 6, but the changes were not significant (*p* > 0.05), *n* = 22. The mean TSL thickness increased from the baseline to month 6, and the increase was significant (*p* < 0.05) for the nine lesion areas that showed measurable TSLs.

## 4. Discussion

The goal of this study was to demonstrate that CP-OCT could monitor changes in the lesion activity and severity over time during active monitoring. We anticipated seeing initially active lesions in the CP-OCT images with no TSLs at the baseline transition to arrested lesions with TSLs at month 6. Unfortunately, only 1 of the 22 lesion sites exhibited that ideal behavior. It is disappointing that most of the lesion sites did not improve with the use of 1.1% NaF, however, it is important to point out that this study was not a clinical trial on the remineralization performance of high fluoride dentifrices, nor could we control the compliance and hygiene of the participants. CP-OCT was able to show changes in many of the lesions over time, and this study demonstrated that those lesion sites that appeared initially arrested with the presence of a measurable TSL did not progress further in lesion severity. Therefore, the presence of a measurable TSL present on the lesion in CP-OCT images appears to be a key indicator that the lesion is arrested and did not progress during the six-month study. CP-OCT detected the presence of a TSL in a single measurement so lesion activity could be assessed at a single time point in vivo. The use of the highly mineralized transparent surface zone as our “benchmark” for an arrested lesion surface is supported by several in vitro and in vivo studies, and the histopathology of active vs. arrested lesions [31,33,34]. Before the introduction of the optical coherence tomography, a tooth had to be extracted, sectioned, and imaged with polarized light microscopy or transverse microradiography to detect and measure the thickness of a transparent surface zone [6,35]. Of those methods, only OCT can be used clinically.

In this study, CP-OCT was also able to monitor changes in the dimensions of the gingival tissues. We did not anticipate that the gingiva in some patients would change so dramatically in 6 months. For example, in Figure 5 the gingival margin appears to advance more than a millimeter from the baseline to 6 months to nearly cover the previously exposed root surface. Such changes are challenging for the image registration since the gingival margin cannot be used as a fixed reference. Since active root lesions are soft, considerable erosion is expected and CP-OCT was able to show that at multiple sites, the demineralized dentin was preferentially eroded/abraded away over time. The severity of the demineralization can be quantified using CP-OCT by the measurement of the lesion depth and the integrated reflectivity over the lesion depth (ΔR). However, this approach is problematic when considerable erosion is present, as can be the case for root lesions. For example, the lesion site in Figure 3 shows large drops in the lesion depth and ΔR after 6 months. This is not due to remineralization but because the softer areas of demineralization were likely abraded/eroded away. The mean lesion depth and ΔR increased slightly over 6 months but neither increase was statistically significant. However, considering the influence of the erosion discussed above, it is likely that, with the exception of the nine lesions that manifested a TSL at 6 months, most of the lesions progressed and increased in severity.

This study and our prior study [27] further demonstrate that, not only is it difficult to visually assess the activity of root caries, but it is also difficult to visually identify whether or not demineralization is present on the exposed root surfaces, or if the lesion is actually located on the root surface. The dentinal enamel junction (DEJ) and the cementum enamel junction (CEJ) are easy to identify in OCT images, but are often very difficult to visually identify, particularly in lesion areas. For many of the lesions encountered in this study, the location of demineralization in OCT images did not correspond to where it was anticipated based on the lesion color and texture. Tactile assessments of root caries lesions are more reliable, however, tactile assessments and aggressive probing with the explorer risks further damage to lesion areas and the use of the sharp explorer is discouraged. It is also important to note that OCT has limited access to subgingival lesion areas.

## 5. Conclusions

This was the first clinical study to show that CP-OCT can be used to monitor changes in lesion structure and activity over time during active monitoring. It demonstrates the ability of CP-OCT to identify and monitor the activity of root caries in adults, and shows the limitations of diagnosis by visual inspection, as compared to CP-OCT. CP-OCT imaging has great potential for lesion activity assessment in a single visit. In addition, CP-OCT was able to monitor the erosion of the root surface, loss of the cementum layer, changes in the gingival tissues, and provide anatomic landmarks of the DEJ and CEJ. The therapeutic relevance of the study is that it shows that CP-OCT can be used to monitor the efficacy of chemical intervention and can potentially be used in clinical trials to establish the effectiveness of potential treatment options for root caries.

## Figures and Tables

**Figure 1 diagnostics-13-00465-f001:**
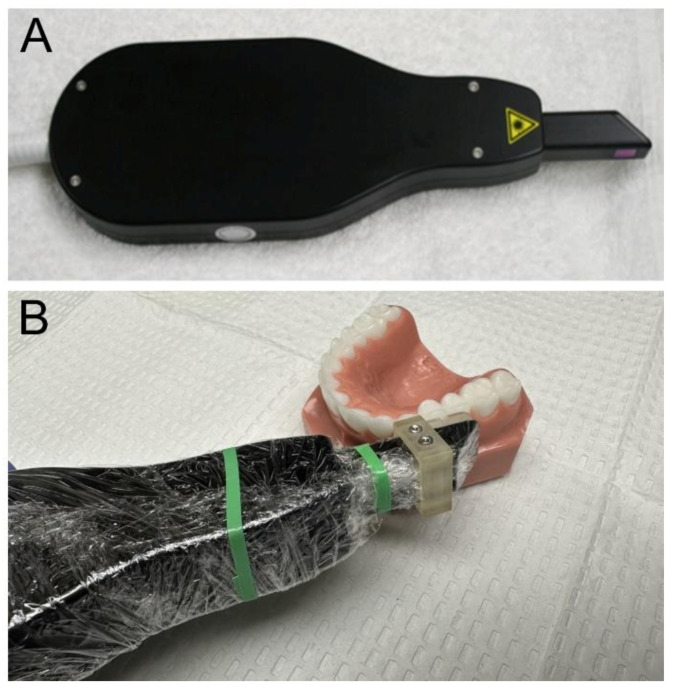
(**A**) CP-OCT handpiece used for these studies, and (**B**) a closeup of the handpiece with the added 3D printed attachment and covered by polypropylene wrap for infection control.

**Figure 2 diagnostics-13-00465-f002:**
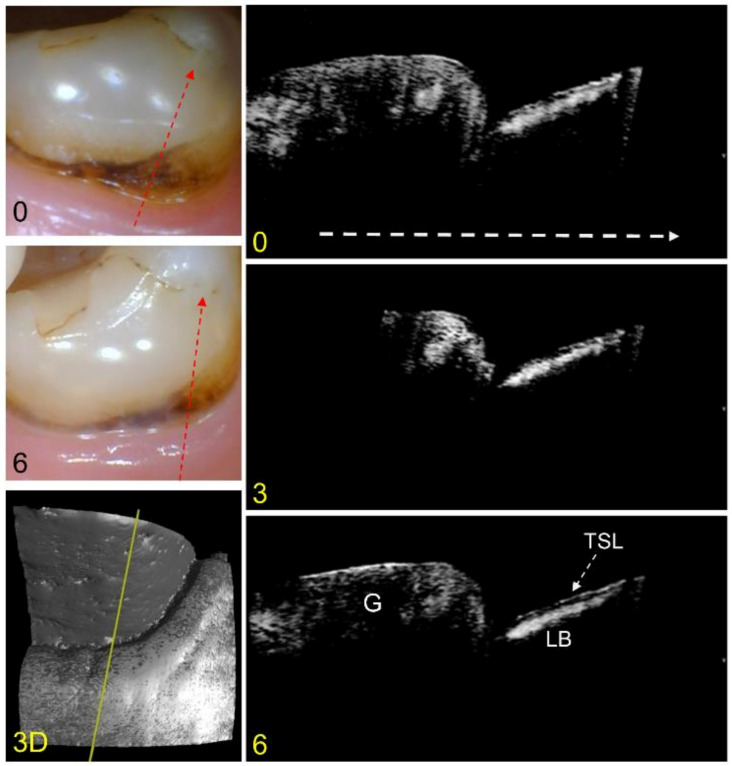
Color pictures at baseline and 6 months are shown for an arrested lesion. The dashed red lines show the likely position of the three CP-OCT b-scans shown at baseline, 3 months, and 6 months. The white dashed arrow in the OCT b-scan is directed from the root to the crown. A surface rendering (3D) of the entire CP-OCT scan taken at baseline is also shown, with the yellow line showing the position of the extracted b-scans. The position of the gingiva (G) and the lesion body (LB) with the TSL located on top of the lesion are indicated. Little change has occurred for the lesion over 6 months.

**Figure 3 diagnostics-13-00465-f003:**
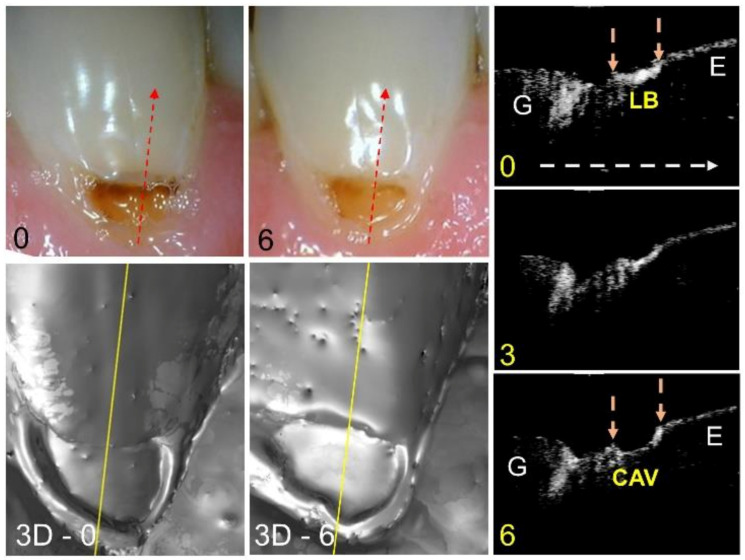
Color pictures at baseline and 6 months are shown for an active lesion with advancing cavitation. The dashed red lines show the likely position of the three CP-OCT b-scans shown at baseline, 3 months, and 6 months. The white dashed arrow in the OCT b-scan is directed from the root to the crown. Surface renderings (3D) of the entire CP-OCT scans taken at baseline and 6 months are also shown, with the yellow lines showing the position of the extracted b-scans. The position of the gingiva (G) and enamel (E) along with the lesion body (LB) are indicated. After 6 months, the demineralized area encompassing the lesion body between the two arrows has eroded, leaving a cavitated region (CAV).

**Figure 4 diagnostics-13-00465-f004:**
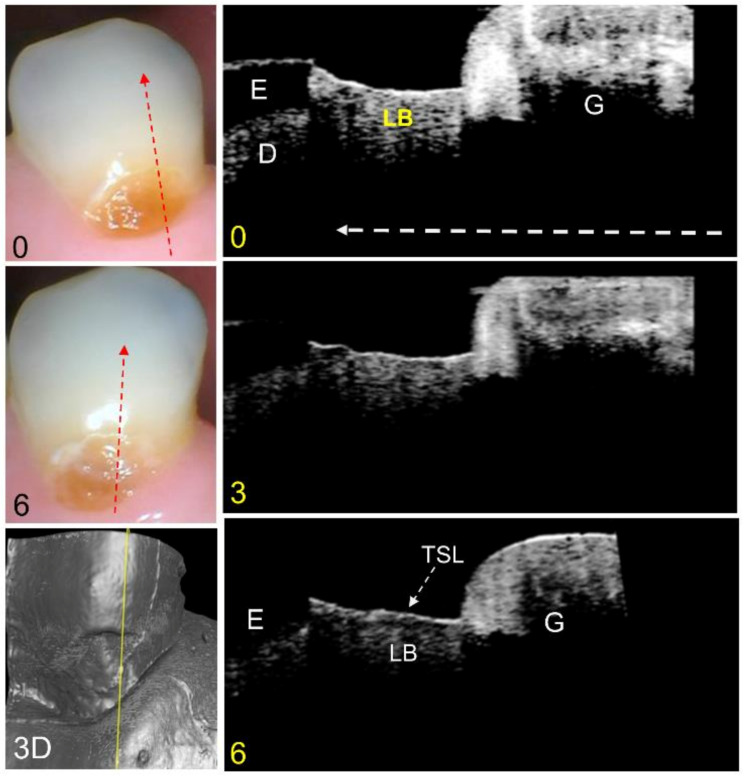
Color pictures at baseline and 6 months are shown for an active lesion that has undergone increasing remineralization and has a TSL after 6 months. The dashed red lines show the likely position of the three CP-OCT b-scans shown at baseline, 3 months, and 6 months. The white dashed arrow in the OCT b-scan is directed from the root to the crown. A surface rendering (3D) of the entire CP-OCT scan taken at 0 months is also shown, with the yellow line showing the position of the extracted b-scans. The position of the gingiva (G), enamel (E), and underlying dentin (D) are shown, and the lesion body (LB) is indicated. After 6 months, the intensity of reflectivity from the lesion body has decreased, and a distinct TSL is visible at the lesion surface.

**Figure 5 diagnostics-13-00465-f005:**
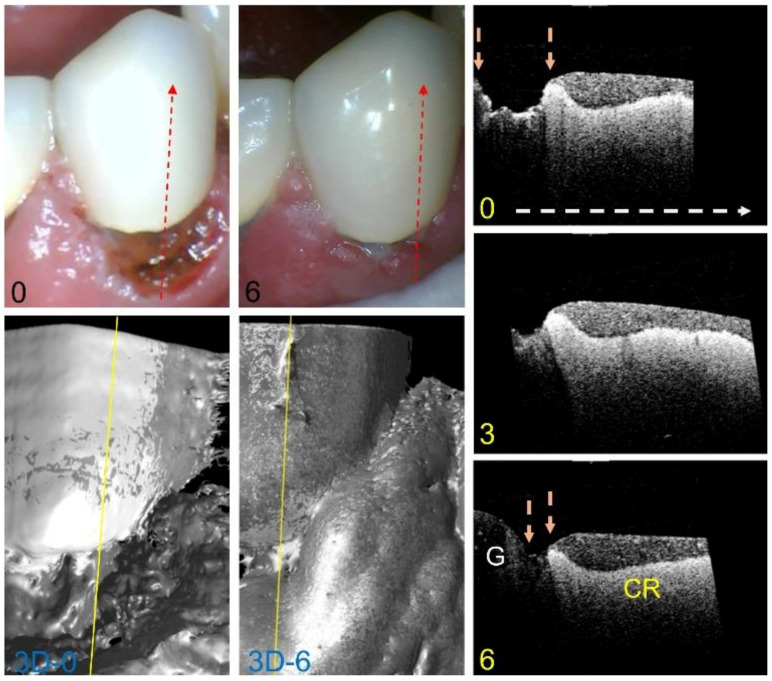
Color pictures at baseline and 6 months of an exposed root surface that becomes completely covered by inflamed gingiva after 6 months. The dashed red lines show the likely position of the three CP-OCT b-scans shown at baseline, 3 months, and 6 months. The white dashed arrow in the OCT b-scan is directed from the root to the crown. Surface renderings (3D) of the entire CP-OCT scans taken at baseline and 6 months are also shown, with the yellow lines showing the position of the extracted b-scans. The position of the gingiva (G) and the attached crown (CR) are indicated. After 6 months, the exposed root surface indicated by the position between the two arrows has contracted markedly and the 3D surface rendering shows most of the root surface that was exposed at baseline is now covered by the expanded gingiva.

**Figure 6 diagnostics-13-00465-f006:**
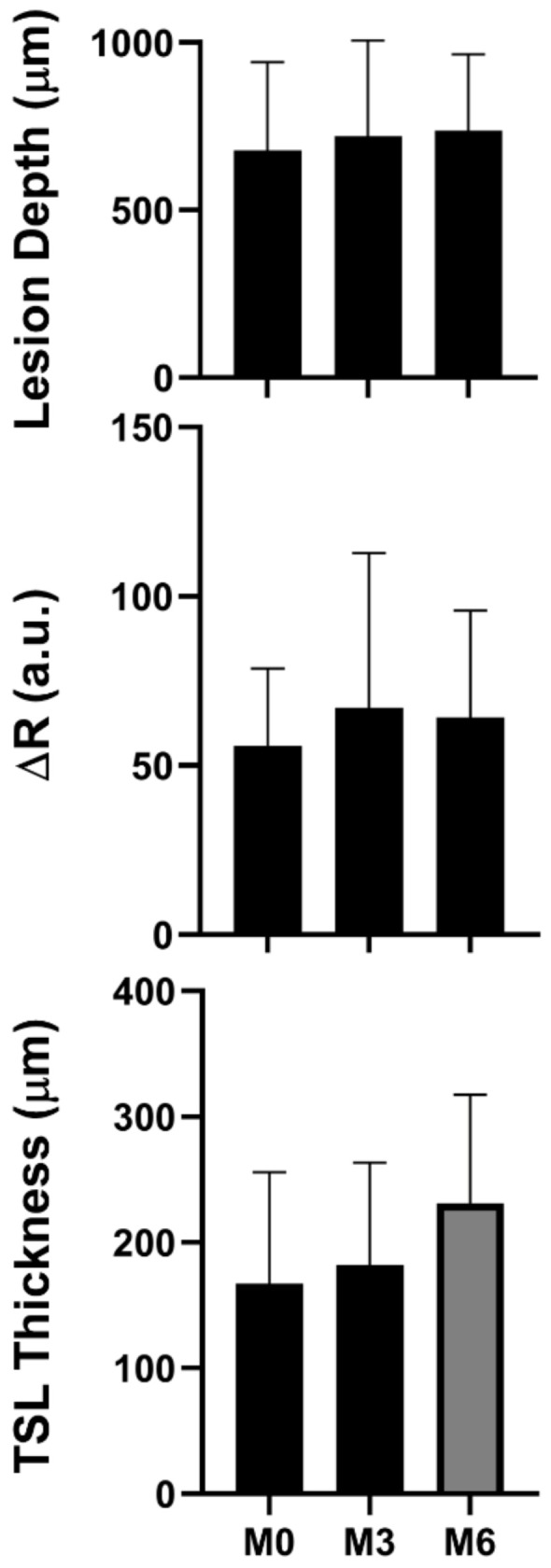
Bar graphs for mean ± S.D. for the CP-OCT measurements of the lesion depth, integrated reflectivity over the lesion depth (ΔR), and the TSL thickness. Bars of the same color are statistically similar (*p* > 0.05). There were 22 samples for the lesion depth and ΔR, and 9 samples that had measurable TSLs.

**Table 1 diagnostics-13-00465-t001:** Contingency table comparing visual assessment of the digital images of the 22 lesion areas to the CP-OCT images at baseline and at 6 months. The diagonal elements (bold) of the contingency table show the number of visual assessments that match the CP-OCT analysis.

		CP-OCT (Baseline)				CP-OCT (Month 6)			
		No Lesion	Active Lesion	Arrested Lesion	Total	No Lesion	Active Lesion	Arrested Lesion	Total
**Visual**	No Lesion	**0**	5	1	6	**0**	6	1	7
	Active Lesion	0	**9**	7	16	0	**5**	8	13
	Arrested Lesion	0	0	**0**	0	0	2	**0**	2
	Total	0	14	8	22	0	13	9	22

## Data Availability

Not applicable.

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
