# Peer review of "Active Surveillance of Root Caries in Vivo with CP-OCT"

_diagnostics, 2023, doi:10.3390/diagnostics13030465_

Round 1

Reviewer 1 Report

This study evaluated the changes in root caries lesion by the method for monitoring root caries using CP-OCT. This is a very interesting study on root caries monitoring, but the following points need to be answered or corrected. This study has some concerns.

Concerns:

1.     Root caries is more extensive than coronal caries and can extend below the gingiva. A line is set for caries detection, but there are parts where it is questionable whether the line in the image is drawn correctly. Since the reference point is unclear, please describe in detail.

2.     Figure 1B The direction shown in the photograph does not seem to allow insertion into the oral cavity, so the image should be revised in consideration of insertion into the oral cavity.

3.     Please explain clearly what the thickness of TSL indicates. If diffuse reflection occurs there, wouldn't it be possible that the surface layer is demineralized?

4.     I think that it is important to supplement the same point as a reference and to have a short time for monitoring. Is superimposition or 3D analysis possible as a solution to the unclear reference point? How long does it take to analyze one caries lesion?

5.     L271 “a mm” should be changed.

Author Response

We have responded to the reviewers’ requests and comments to the best of our ability.  Our detailed response is indicated in italics.  We greatly appreciate the efforts of the reviewers to improve the manuscript and we hope we have sufficiently addressed their concerns.

Review 1

This study evaluated the changes in root caries lesion by the method for monitoring root caries using CP-OCT. This is a very interesting study on root caries monitoring, but the following points need to be answered or corrected. This study has some concerns.

Concerns:

  1. Root caries is more extensive than coronal caries and can extend below the gingiva. A line is set for caries detection, but there are parts where it is questionable whether the line in the image is drawn correctly. Since the reference point is unclear, please describe in detail.

- We have added to the discussion:  It is important to note that OCT has limited access to subgingival lesion areas.

The lines drawn in the color images are there to indicate the approximate position where the CP-OCT slices were extracted from the 3D CP-OCT image.  Lines are also shown on the 3D CP-OCT images.   The three CP-OCT images were co-registered using an image registration algorithm in ORS Dragonfly.  Visible/color images cannot be co-registered.  We have redrawn the lines in the color images of Figures 2 and 5 to better match the positions in the CP-OCT images.

  1. Figure 1B The direction shown in the photograph does not seem to allow insertion into the oral cavity, so the image should be revised in consideration of insertion into the oral cavity.

- Figure 1B has been modified to better represent the imaging position on root surfaces.

  1. Please explain clearly what the thickness of TSL indicates. If diffuse reflection occurs there, wouldn't it be possible that the surface layer is demineralized?

-The transparent surface layer (TSL) thickness is the thickness of the highly mineralized surface layer above the lesion body that is present on arrested lesions.   The surface layer is more transparent, i.e. has a lower reflectivity, than the lesion body in the CP-OCT image.  When minerals are deposited in the outer layers of the porous lesion, the mineral fills the pores reducing scattering and reflectivity and those layers become more transparent.  Several studies utilizing optical coherence tomography, transverse microradiography, microcomputed tomography and polarized light microscopy have confirmed the nature of the transparent surface layer (TSL), See refs. 17, 19, 25-27 and 32-35.    For dentin the TSL will likely have a higher mineral content than sound dentin since the tubules are filled with mineral.  It is this outer zone of high mineral content that inhibits the diffusion of fluids into and out of the lesion and “arrests” the lesion.  The detection of the presence of a TSL is a key indicator of lesion arrest.  The TSL in Figure 2 has a lower reflectivity than the newly formed TSL in Figure 4 while no TSL is visible in the active lesion of Figure 3.

Reflection from the tooth surface is specular reflectance and that reflection is reduced by 2-3 orders of magnitude since this is a cross polarization OCT system.  The reflectivity from deeper layers including both the transparent surface layer and the lesion body is diffuse reflection and that reflectivity increases with increasing demineralization.  The thin layer on top of the TSL is the specular reflectance from the tooth surface.  If we had polarizers of infinite extinction ratio “perfect polarizers” the surface reflection above the TSL would not be visible and we would not be able to measure the thickness of the TSL.  A PS-OCT system would be needed that acquired both cross polarization and co-polarization images (see Fig. 2 reference 34).

  1. I think that it is important to supplement the same point as a reference and to have a short time for monitoring. Is superimposition or 3D analysis possible as a solution to the unclear reference point? How long does it take to analyze one caries lesion?

-The three 3D CP-OCT images for each lesion were aligned or co-registered in ORS Dragonfly using the sum of squared differences method. 

The CP-OCT system acquires 3D images 6x6x7-mm in 3 seconds.  During imaging the individual parallel b-scans are shown in sequence as they are acquired so that important features can be seen in real-time with no post-processing needed.   For example, the clinician scanning the lesion shown in Figure 2 would be able to see the b-scans shown in real-time as they were acquired and identify that a TSL is present on the lesion and that it is most likely arrested without any postprocessing. 

L271 “a mm” should be changed.

-corrected

Reviewer 2 Report

The study aims to determine if cross-polarization optical coherence tomography (CP-OCT) could detect changes in lesion severity and activity during active monitoring. The authors present concise yet informative research. They guide the reader through the research process and explain different aspects of the study. Nevertheless, I advise highlighting the study’s innovation and novelty compared to previous papers.  

Author Response

We have responded to the reviewers’ requests and comments to the best of our ability.  Our detailed response is indicated in italics.  We greatly appreciate the efforts of the reviewers to improve the manuscript and we hope we have sufficiently addressed their concerns.

Review 2

The study aims to determine if cross-polarization optical coherence tomography (CP-OCT) could detect changes in lesion severity and activity during active monitoring. The authors present concise yet informative research. They guide the reader through the research process and explain different aspects of the study. Nevertheless, I advise highlighting the study’s innovation and novelty compared to previous papers.  

-Previous in vitro studies employing CP-OCT and PS-OCT have shown that a TSL forms at the surface of lesions on enamel and dentin surfaces when active lesions become arrested and the thickness of that TSL can be measured nondestructively using CP-OCT.  Other studies employing time resolved thermal imaging and short wavelength infrared imaging to measure the lesion dehydration dynamics have confirmed that the lesion activity can be correlated with the TSL thickness measured with CP-OCT.  A recent clinical study (ref. 27) confirmed that CP-OCT can be used to measure TSL presence to assess lesion activity.  In that study lesions were imaged at a single time point.  The purpose of this study was to show that CP-OCT can be used to show changes in lesion structure and activity over time during active monitoring.

Changes: added at a single time point to L 61 in introduction

Conclusion added sentence:  This was the first clinical study to show that CP-OCT can be used to monitor changes in lesion structure and activity over time during active monitoring.

Reviewer 3 Report

Dear authors, 

1. please explain the power analysis for the sample calculation

2. the age range for the tooth sample is too wide, the biology is different for young permanent teeth when compared to older one, as well as progression of the caries lesions

3. conclusion should be modified and try to include the potential clinical implications, not just for the diagnosis but also from the therapeutic point of view and potential treatment options

4. since you have more then 45 references, discussion should ne re-written or reference list should be reduced

Author Response

We have responded to the reviewers’ requests and comments to the best of our ability.  Our detailed response is indicated in italics.  We greatly appreciate the efforts of the reviewers to improve the manuscript and we hope we have sufficiently addressed their concerns.

Review 3

  1. Please explain the power analysis for the sample calculation

-Added sample size calculation: In an in vitro study of the remineralization of simulated dentin lesions, the mean transparent surface layer thickness (TSL) increased from 0 to 53.8±13.3 µm after 12 days of remineralization [31].  Therefore, based on an average SD of 6.65-µm for TSL thickness with a 2-sided significance level of 0.05, a study with 30 subjects would have a 90% power to detect an increase in TSL thickness of 5.7 µm.

  1. The age range for the tooth sample is too wide, the biology is different for young permanent teeth when compared to older one, as well as progression of the caries lesions

-We agree that “the biology is different for young permanent teeth when compared to older one” and wanted to include this range since root caries is not confined to just the elderly population as indicated below.  

In the National Survey of Adult Oral Health in Australia 2004-06 (NSAOH 2004-06), the prevalence of untreated root caries was 6.7% in all adults 15+ years old, but increased to 7.1% in 35- to 54-year-olds, 12.6% in 55- to 74-year-olds and 17.3% in those aged 75+ years old. (Hariyani, N  Root caries experience among Australian adults, Gerodontology, 2017) 

  1. Conclusion should be modified and try to include the potential clinical implications, not just for the diagnosis but also from the therapeutic point of view and potential treatment options

- This was the first clinical study to show that CP-OCT can be used to monitor changes in lesion structure and activity over time during active monitoring. It demonstrates the ability of CP-OCT to identify and monitor the activity of root caries in adults and shows the limitations of diagnosis by visual inspection as compared to CP-OCT.  CP-OCT imaging has great potential for lesion activity assessment in a single visit.  In addition, CP-OCT was able to monitor erosion of the root surface, loss of the cementum layer, changes in the gingival tissues, and provide anatomic landmarks of the DEJ and CEJ.  The therapeutic relevance of the study is that it shows that CP-OCT can be used to monitor the efficacy of chemical intervention and can potentially be used in clinical trials to establish the effectiveness of potential treatment options for root caries. 

  1. Since you have more then 45 references, discussion should be re-written or reference list should be reduced

-Removed redundant references and there are now 35 references

Round 2

Reviewer 1 Report

Thank you for your answer and correction. Please answer the following concerns.

Concerns:

1.     I understand that remineralization of root caries is complicated, but is it possible to say that TSL by OCT is not affected by metal deposition by high concentrations of fluoride, SDF, etc.?

2.     I'm a little unclear about "The three CP-OCT images were co-registered using an image registration algorithm in ORS Dragonfly." in your answer. Please be specific about whether the image is distorted, shrunk or enlarged, and whether it is automatic or intentional.

Author Response

  1. I understand that remineralization of root caries is complicated, but is it possible to say that TSL by OCT is not affected by metal deposition by high concentrations of fluoride, SDF, etc.?

Resolution of the TSL in OCT images is not likely influenced by high fluoride concentrations.  In remineralization studies the TSL is produced using supersaturated solutions with added fluoride to enhance the deposition of fluoroapatite so most TSLs that have been imaged OCT likely have a high fluoride content.  In addition, the TSLs are likely more resistant to further demineralization during a future acid challenge due to the lower solubility compared to sound enamel or dentin.

SDF is more complicated.  We have investigated the influence of SDF on OCT imaging in vitro on enamel and dentin active lesions with no TSL.  Silver deposition in the lesion pores and precipitation on the surface raises the reflectivity and reduces the optical penetration depth.  However, OCT was still able to image through the simulated lesions and since TSLs are located near the surface we would expect to be able to image any existing TSLs on arrested lesions.  Moreover, the reflectivity returned to normal levels in OCT images 12 weeks after SDF treatment suggesting that the silver precipitates are eventually leached out of the lesion.  

Abdelaziz M, Yang  V, Chang  NN, Darling CL, Fried WA, Seto  J, Fried D, Monitoring Silver Diamine Fluoride Application with Optical Coherence Tomography and Thermal Imaging: An In Vitro Proof of Concept Study,  Lasers Surg Med, 53(7) 968-977 (2022).

Therefore, we can hypothesize that any TSLs that form after SDF treatment would be readily visible after 6 months.  We hope to carry out a future study in which we monitor root caries lesions treated with SDF with OCT for 6 months after application.

  1. I'm a little unclear about "The three CP-OCT images were co-registered using an image registration algorithm in ORS Dragonfly." in your answer. Please be specific about whether the image is distorted, shrunk or enlarged, and whether it is automatic or intentional.

Registration was carried out only by rotation and translation of the three OCT  images in 3D.  With OCT any distortion between images is likely due to the fact that OCT measures optical path length and not physical path length.  The optical path length is influenced by the index of refraction of the medium that is higher for dentin and enamel compared to air and water.  Therefore, since the images are obtained at different angles there is always some distortion due to the original optical path differences.

Round 3

Reviewer 1 Report

I have some doubts about the reproducibility of OCT, but since the authors do not seem to be sincere about the description of the experimental method for processing OCT images, I accept it.

Author Response

I have some doubts about the reproducibility of OCT, but since the authors do not seem to be sincere about the description of the experimental method for processing OCT images, I accept it.

We were able to co-register most of the OCT images at the three time points, but the reviewer is correct that the images are not perfectly matched.  Co-registration minimizes the differences in the images based on image rotation and translation but does not ensure a perfect match. Tissues such as the gingiva change in dimension over time.  Additionally, with any handheld imaging device, changes in distance and angle from the tissue surface can slightly change the 3D image acquired.  OCT is similar in nature to ultrasound.  In OCT any changes in refractive index change the optical path length in the image while in ultrasound any changes in tissue density change the sound path length.  Therefore, changes in angle can “distort” the images.  

We assume that clinicians would be most interested in examining the lesions to see if the lesions have progressed and are more severe or a transparent surface zone has formed indicating that they have become arrested.  Such assessments do not require overly precise measurements or the need for micron level image registration to make such assessments.  Therefore, we strongly believe the images are sufficiently co-registered in this study to demonstrate the utility of OCT for the clinical monitoring of root caries lesions.  

In response to the reviewer suggestions, we have added additional information regarding the image registration to the methods section.

Image preprocessing was first applied using ORS Dragonfly (Montreal, Canada) to despeckle and smooth the images before image registration [31,32]. Image segmentation was first performed with 90% intensity thresholding, followed by application of a 3D median filter with a kernel size 3 for noise reduction and smoothing while preserving the potential transparent surface layer (TSL).  Image registration was first performed using manual rotation and translation to achieve an initial rough alignment. It was followed by application of ORS Dragonfly’s automatic image registration function.   It automatically tests the registration by introducing miniature steps of rotation as small as 0.5 degrees and translation of 1 voxel with a linear type of interpolation until the mobile image is superimposed to the fixed image.  Alignment is assessed by the Sum of Squared Differences (SSD) method.  It is important to note that OCT measures optical path length and not physical path length, so some distortion is expected for images acquired at different angles.